# Effects of Exposure to Low Zearalenone Concentrations Close to the EU Recommended Value on Weaned Piglets’ Colon

**DOI:** 10.3390/toxins15030206

**Published:** 2023-03-09

**Authors:** Iulian Alexandru Grosu, Cristina Valeria Bulgaru, Gina Cecilia Pistol, Ana Cismileanu, Daniela Eliza Marin, Ionelia Taranu

**Affiliations:** National Research and Development Institute for Biology and Animal Nutrition IBNA, Calea Bucuresti No. 1, 077015 Balotesti, Romaniagina.pistol@ibna.ro (G.C.P.); ana.cismileanu@ibna.ro (A.C.); daniela.marin@ibna.ro (D.E.M.); ionelia.taranu@ibna.ro (I.T.)

**Keywords:** piglets, zearalenone, microbiota, short chain fatty acids

## Abstract

Pigs are the most sensitive animal to zearalenone (ZEN) contamination, especially after weaning, with acute deleterious effects on different health parameters. Although recommendations not to exceed 100 µg/kg in piglets feed exists (2006/576/EC), there are no clear regulations concerning the maximum limit in feed for piglets, which means that more investigations are necessary to establish a guidance value. Due to these reasons, the present study aims to investigate if ZEN, at a concentration lower than the EC recommendation for piglets, might affect the microbiota or induce changes in SCFA synthesis and can trigger modifications of nutritional, physiological, and immunological markers in the colon (intestinal integrity through junction protein analysis and local immunity through IgA production). Consequently, the effect of two concentrations of zearalenone were tested, one below the limit recommended by the EC (75 µg/kg) and a higher one (290 µg/kg) for comparison reasons. Although exposure to contaminated feed with 75 µg ZEN/kg feed did not significantly affect the observed parameters, the 290 µg/kg feed altered several microbiota population abundances and the secretory IgA levels. The obtained results contribute to a better understanding of the adverse effects that ZEN can have in the colon of young pigs in a dose-dependent manner.

## 1. Introduction

Zearalenone (ZEN) is an estrogenic nonsteroidal mycotoxin which contaminates cereal grains and their derived products [1,2]. Multiple *Fusarium* species, including *F. graminearum* and *F. culmorum* produce ZEN that presents a significant risk for the livestock industry and an ongoing threat to human and animal health [3,4]. Sensitivity to ZEN varies according to species, swine being reported as the most sensitive among the livestock animals, 1.1 mg ZEN/kg feed for 18 days being enough to induce reproductive and endocrinal disorders in post-weaning gilts, affecting their ovarian follicles in a dose-related manner [5]. A survey of the last nine years reported by the Austrian Biomin Company shows that ZEN was detected in 40–71% of swine feed samples [6]. However, at the level of the European Commission, there are no regulations regarding the maximum limit allowed in the feed for young pigs, though only recommendations, meaning that additional studies are necessary to establish a precise regulation with this limit. For example, the effects of ZEN on the pig gut microbiota have not yet been sufficiently studied, especially at concentrations lower than 100 ppb [7]. A stable microbiome with well-balanced bacterial populations is necessary to maintain normal body function and prevent symptoms such as dysbiosis [8]. Generally, and especially in young pigs after weaning, gut microbiota plays a pivotal role in stimulating the immune system development, facilitating nutrient absorption, and preventing pathogen colonization through competitive exclusion. Carbohydrates and other macronutrients the host cannot assimilate are fermented under anaerobic conditions by gastrointestinal microorganisms resulting in the production of short-chain fatty acids (SCFAs) such as acetic acid, propionic acid, and butyric acid with essential roles in the immune and metabolic systems and maintenance of intestinal barrier function [7,9,10]. Under normal conditions, there is a permanent exchange of information between intestinal epithelial cells and gastrointestinal microbiota, indicating that the gut microbiota is essential in the development and function of the gastrointestinal tract [5,11]. Perturbations in the gut microbiota have been involved in the triggering or as a contributor to many diseases [12]. Exposure of animals to mycotoxins lead to modifications of the gut microbiome composition at the species, phylum, and genus level [7,12]. Such modifications may be caused by the antibacterial mycotoxins effect or by the mycotoxins effect on immune cells and epithelium, on the amount and composition of the volatile fatty acids (SCFAs) and/or on other microbial metabolites in the gut [2,13]. In mice, for example, the deterioration of the immune response was registered following exposure to ZEN at 48.5 mg/kg feed, along with an alteration of the intestinal barrier and villous functions [14]. Rodents are known to have some of the lowest α-ZOL metabolite concentrations produced by their liver in the process of ZEN biotransformation. For this reason, in rodents, the deleterious effects appear only after the exposure to high doses of ZEN [15]. In cows, it was found that 20 mg ZEN /kg feed reduced the rumen pH and the concentration of total SCFAs [16]. Compared to monogastric animals, ruminants are significantly less sensitive to ZEN exposure as microbial rumen flora play a role in ZEN elimination [17]. Pigs are very sensitive to ZEN toxicity, the most critical effect of ZEN being on the reproductive system [10,11]. However, the mycotoxin effects on the gastrointestinal tract cannot be neglected as the mycotoxin impairs the viability of intestinal cells, lowers the trans-epithelial protection and the synthesis of cytokines, affecting thus the health of the gastrointestinal tract [18,19]. After oral administration, ZEN is quickly absorbed in the gastrointestinal tract, and evidence shows significant effects of ZEN at this level [12,20]. Furthermore, it was shown by biotransformation studies conducted on sub-cellular fractions of pig liver that the primary hepatic metabolite of ZEN is α-zearalenol (α-ZOL) [21]. The authors also detected extrahepatic biotransformation of ZEN into α-ZOL within the porcine granulosa cells [22]. The high sensitivity of pigs towards ZEN could be explained by the fact that α-ZOL binds more reliably than ZEN to the nuclear receptors within the liver.

The gastrointestinal tract assures the defence against xenobiotic agents, including mycotoxins, constituting a physical barrier in that sense. Intestinal epithelial cells (IECs) are an integral part of this barrier [23], and are essential components associated with the host’s innate and adaptive immunity [2]. The intestinal barrier function is assured by membrane proteins constituting the tight junction network within intestinal epithelial cells [18,24]. These junctions seal the gut lumen and limit the transport of small hydrophilic molecules [25]. However, recent studies have shown that ZEN induced intestinal barrier dysfunction by decreasing the mRNA expression of claudin-1, occludin, claudin-3, and ZO-1 tight junction proteins along with the redistribution of ZO-1 immunofluorescence [26].

Considering the above, the present study investigated if zearalenone at concentrations lower than the EC recommendation (no. 576/2006, 100 ppb) might affect the colon microbiota and other markers relevant to nutritional, physiological, and immunological status for growing pigs. For this purpose and to serve as a comparison, the effect of two concentrations of ZEN were tested, one below the limit recommended by the EC (75 ppb) and the other higher for comparison (290 ppb).

## 2. Results

### 2.1. Effect of ZEN on the Percentage of Animals with Diarrhoea and on Animal Performances

During the experimental period, the piglets presenting diarrhoea symptoms were registered for each experimental group, and the results are presented in Figure 1. Exposure to zearalenone has increased the number of animals with diarrhoea, directly proportional to the concentration of ZEN (37.5% piglets with diarrhoea in the ZEN 75 group, respectively, 50% in the ZEN 290 group versus only 12.5% in the Control group).

Although slight differences between experimental groups could be observed for the growth parameters, these differences were not significant (Table 1). The final body weight of piglets from the ZEN 75 and ZEN 290 group was lower compared to the Control. Slight decreases in ADG and ADFI could also be observed in ZEN experimental groups when compared with the Control group, but they were not significant. As such, ADG and ADFI parameters were similar for ZEN 75 and ZEN 290 (0.377 and 0.370 kg for ADG and 1.038 and 1.042 kg for ADFI) and were lower than the values in the Control group (0.424 kg and 1.144 kg). The best feed efficiency was observed for the Control group with a 2.6 ratio, compared to 2.77 for the ZEN 75 group and 2.82 for the ZEN 290 group; however, the differences were not significant.

### 2.2. Relative Abundance of Several Important Microbial Populations under the Influence of ZEN

Five bacterial genera were chosen as markers (*Lactobacillus, Bifidobacterium, Prevotella, Enterobacter,* and *Clostridium*) for a healthy intestinal tract based on the scientific literature and our previous studies. Their corresponding populations were determined from samples of colonic content (Figure 2) and calculated as relative abundances. At 75 ppb, ZEN significantly influenced only the *Lactobacillus* population lowering it by 61% (*p* < 0.05, Figure 2), while not affecting *Prevotella, Clostridium, Enterobacter,* and *Bifidobacterium* when compared to control (Figure 2). By contrast, *Prevotella, Clostridium, and Enterobacter* relative abundances were significantly impacted by the presence of ZEN at 290 ppb, registering an increase over the Control group and ZEN 75 group. The most spectacular change was observed for the *Enterobacter* population, increasing 50 times over that of the Control group (*p* < 0.05) (Figure 2), followed by *Clostridium*, with an increase of 17.7 times (*p* < 0.05) and *Prevotella* with the lowest abundance increase, of only eight times over the Control group (*p* < 0.05) (Figure 2). Similar to the ZEN 75 group, ZEN at 290 ppb decreased the abundance of the *Lactobacillus* population by over 60% compared to the Control (Figure 2). *Bifidobacterium* relative abundance was also significantly lowered, registering a decrease of 6.18% when compared with the Control group (*p* < 0.05) (Figure 2). Similar to the ZEN 75 group, ZEN at 290 ppb significantly decreased the abundance of the *Lactobacillus* population by over 60% compared to the Control (Figure 2). In addition, *Bifidobacterium* relative abundance was also significantly lowered, registering a decrease of 6.18% when compared with the Control group (*p* < 0.05) (Figure 2).

### 2.3. Effect of ZEN on the SCFA Colonic Content, pH Value and Ammonia Concentration

The SCFA concentrations among the experimental groups are presented in Table 2. The levels of acetic, propionic, isobutyric, butyric, isovaleric and valeric acids as well as total short-chained fatty acids did not significantly differ between the experimental groups for ZEN 75 or ZEN 290 when compared to the control. The highest SCFA concentrations were observed for acetic, propionic and butyric acid, irrespective of treatment. On the opposite side, isovaleric and isobutyric acids had the lowest observed values. The total concentration of SCFAs was slightly higher in the ZEN 290 group due to elevated concentrations of acetic acid, and propionic acid observed within this group but not significant.

ZEN presence within the diet did not alter in any significant way the pH values or ammonia concentrations between the experimental groups.

### 2.4. Correlations between SCFAs and Selected Microbial Populations

The effect of ZEN concentrations on the experimental groups was further expanded by Pearson’s correlations between microbial species and SCFA production within the colon, presented in Table 3. The data were also represented in three heatmaps, corresponding to the Control group that received the standard diet and the ZEN dietary groups with differing ZEN concentrations (Figure 3). Positive correlations pointed to a high probability of association, and a negative correlation was related to an absence of a relationship rather than an inhibitory process. The calculated correlations for the Control group between bacteria and SCFAs revealed that *Lactobacillus* was strongly associated with valeric acid (*p* = 0.0214) and moderately associated with isobutyric acid (*p* = 0.0399) within the colon (Table 3, Figure 3a). Additionally, the *Clostridium* genus was highly correlated with propionic acid (*p* = 0.0288) within the colon of piglets unaffected by ZEN. Several positive associations were observed for each microbial genus at different ZEN concentrations (Table 3, Figure 3). At 75 ppb ZEN concentration, both *Lactobacillus* (*p* = 0.0353) and *Bifidobacterium* (*p* = 0.018) were highly correlated with the presence of butyrates (Table 3, Figure 3b). Other strong positive correlations were observed between *Bifidobacterium* and isovalerates (*p* = 0.0189). Moderate correlations were observed between *Enterobacter* and both acetate (*p* = 0.0439) and propionate (*p* = 0.0346), as well as between *Prevotella* and propionates (*p* = 0.0156). The SCFA landscape changed at 290 ppb ZEN (Table 3, Figure 3c). *Lactobacillus* (*p* = 0.0155) and *Bifidobacterium* (*p* = 0.0244) maintained correlations with butyrates. Similarly, to its previous pattern, *Enterobacter* maintained a moderate correlation with acetates (*p* = 0.0144) and propionates (*p* = 0.0416). Strongly negative correlations were also observed between acetic acid, *Bifidobacterium* (*p* = 0.0124), and *Lactobacillus* (*p* = 0.0267). Additionally, *Bifidobacterium* was moderately negatively correlated with propionic acid (*p* = 0.0302) (Table 3, Figure 3c).

### 2.5. Effect of Zearalenone on Secretory IgA Synthesis

In order to evaluate the effect of ZEN on the secretory IgA (sIgA) contents in the intestinal mucosa, we have analysed the IgA levels in the intestinal content (Figure 4). The results showed that exposure of piglets to 290 ppb ZEN can significantly increase the concentration of sIgA in the colonic content (1175 ng/mL versus 789 ng/m in the control group). In contrast, no significant effect on sIgA concentration was observed for the ZEN 75 piglets group.

### 2.6. The Effect of Dietary Zearalenone on Tight Junction Proteins

The present study assessed the effect of ZEN at 75 ppb and 290 ppb concentrations on tight junction proteins at both gene and protein expression. No significant effect was observed for the ZEN 75 group or ZEN 290. In the case of gene expression, a very slight increase was observed for ZO1 at 75 ppb ZEN and 290 ppb but without statistical significance (Figure 5). 

Concerning protein expression, the western blot analysis for all the analysed markers validated the qPCR results, did not significallmt;ly affect with either of the two ZEN concentrations significantly affecting the junction proteins Cldn 4, ZO1 or Occldn (Figure 6).

### 2.7. Microbial Community Clustering and Relationship between

Clusterization of the selected microbial abundances data was checked by employing the Partial Least Squares discriminant analysis (PLS-DA) by regressing on many sample features (e.g., relative abundances) and finding the set of latent, orthogonal factors that help with cluster separation. It is often employed in -omics relative abundances, for example, to assess the discriminatory ability of experimental groups associated with a disease, toxin or phenotype. As shown in Figure 7, there was a clear distinction between experimental groups affected by ZEN and control in their microbiological composition. The differences were more significant between the ZEN 290 group and the rest, this group being the most distant from either the Control or ZEN 75 group.

To investigate any other existing associations and to better visualize the data obtained, another PLS-DA analysis was performed on the following sets of parameters: SCFAs concentrations, microbial genera, pH value, and NH_3_ content with the frame of the experimental groups’ vectors (Figure 8). ZEN at 75 ppb was strongly associated with the isobutyric and isovaleric volatile acids. The pH, acetic acid concentration, and relative abundances of *Clostridium* and *Enterobacter* formed a cluster strongly associated with the presence of ZEN at 290 ppb and moderately associated with the propionic acid and ammonia concentrations.

## 3. Discussion

ZEN, a commonly found contaminant within animal feed, has been widely studied for its toxic effects in swine [2,3,27,28,29]. Comparatively few studies have investigated the effect of ZEN at concentration of 0.100 mg ZEN/kg feed (the limit of EC recommendation) or lower in pig [27]. According to existing studies, the effects of piglets’ exposure to low concentrations of ZEN, under 0.1 ppm, proved challenging to predict [30]. Time of exposure and the quantities of ingested feed are contributing factors that increase the prognosis difficulty. Therefore, even low concentrations of ZEN can induce unexpected responses; the host self-defence mechanisms might ignore the presence of mycotoxin, and the deleterious effects slowly accumulate [30,31]. It is also well known that the diet shift after weaning leading to a second change in bacterial populations. As such, feed quality must be given great importance [20,32]. For example, contaminants in feed, such as ZEN within the intestinal tract can alter microbial ecology homeostasis [7]. If mycotoxin contamination is associated with different pathogens infection as *E. coli*, *Salmonella,* and *Rotaviruses* (due to an immature, emerging immune system), this can lead to the apparition of dysbiosis, intestinal oxidative stress, inflammation, and diarrhoea [33,34].

In the present study, we investigated the effect of a low concentration of ZEN on microbial abundance and other associated markers such as SCFAs, IgA production, pH, ammonia content, and intestinal integrity through junction proteins in the colonic content and colon tissue of piglets fed a diet contaminated with 75 ppb of ZEN (level under the EC recommendation). These parameters were chosen to consider that gut microbiota plays an essential role in developing the immune and digestive systems facilitating nutrient absorption and preventing pathogen colonisation. It also binds ingested toxins and produces antimicrobial molecules and vitamins [35,36]. Within the intestinal tract, the microbiota improves the anti-inflammatory response as well as maintains the intestinal barrier function [36,37]. In addition, SCFAs, tight junction proteins and IgA production are crucial for maintaining intestinal barrier integrity, defence response and gastrointestinal health. In tandem and for comparison reasons, a higher level of ZEN, 290 ppb, above the EC recommendations was also analysed.

Our results showed that at 75 ppb (0.075 mg/kg feed) concentration, ZEN significantly (*p* < 0.05) reduced the *Lactobacillus* population within the colon, with the *Bifidobacterium* population lowered as well, but not in a significant way. A more severe alteration was registered for the 290 ppb (0.290 mg/kg feed) concentration, which decreased both *Lactobacillus* and *Bifidobacterium* significantly while noticeably (*p* < 0.05) increasing the populations of *Prevotella*, *Clostridium,* and *Enterobacter* genera. Although the *Clostridium* and *Enterobacter* genera include well-known pathogens, many of the members belonging to these genera are regular bacterial species representative of a healthy pig microbiota, as pointed out by many other studies [38,39,40]. Nevertheless, our findings show an increase in *Clostridium* and *Enterobacter* populations in the presence of ZEN 290 ppb. This was observed for *Clostridium difficile* in gilts by Cieplinska et al. [7] in challenges with ZEN ranging from 5 to 15 µg ZEN/kg b.w. Le Sciellour et al. [3] similarly reported an increase in *Clostridium* genera in the presence of both ZEN (0.75 mg/kg feed) and DON (3.02 mg/kg feed) in an acute seven-day challenge in female finishing pigs. In a study by Wang et al., [13] an increase in the *Enterobacter* population was reported, along with *Escherichia, Bacteroides, and Salmonella*, in the presence of ZEN. These opportunistic strains can lead to intestinal inflammations in piglets and generate substantial losses in commercial farms [41]. Despite evidence showing the capacity of *Lactobacillus*, along with other lactic acid bacteria such as *Bifidobacterium*, for mycotoxin absorption and biotransformation [12,42] certain studies observed that zearalenone could affect these populations, inducing a decline in their abundance. Li et al. 2018 [43] reported a noticeable decrease in the *Lactobacillus* population at a concentration of 400 µg ZEN/kg b.w. in rabbits in a 28 days challenge. Tan et al. [44] showed a 35% decline in *Lactobacillus* populations at a much lower concentration (40 µg ZEN/kg b.w.) along with a significant change in bacterial populations in mice. A reduction in *Lactobacillus* abundance was also observed in a study conducted on broilers by Jia et al., [4] this effect is correlated with their binding capacity. Moreover, Saenz et al. [45] observed a decrease in the whole taxonomic order of *Lactobacillales* in a 28-day ZEN (1623 ppb) challenge of piglets while observing an increase in the *Bifidobacteriaceae*. The Lactic Acid Bacteria complex, of which *Lactobacillus* and *Bifidobacterium* are members, usually plays a crucial role by producing many SCFAs important to the host and the microbial ecosystem [46]. Being a product of microbial fermentation of refractory carbohydrates, SCFAs are important for maintaining intestinal homeostasis, providing energy to colonocytes and preventing other intestinal diseases [47]. In piglets, for example, propionate is involved in the liver gluconeogenesis process, while acetate is a precursor in synthesising more complex fatty acids [48]. Of the SCFAs, butyrate is important in reducing intestinal inflammation and diminishing systemic infections, acting as an important bacterial and metabolic modulator [49,50]. The health of gastrointestinal tract tissue, mineral absorption and inflammation can also be influenced by normal concentrations of SCFAs [49]. In the present study, acetic acid was found in the most abundant quantity, followed by propionic acid, throughout all the experimental groups. As mentioned above, SCFAs are relevant for the microbial population in the gut. For instance, SCFAs can directly reduce the growth of *Enterobacter* by lowering the pH [25]. They also can reduce the colonisation of *Enterobacter* by suppressing inflammations [51].

These observations could explain why the decrease in *Lactobacillus* and *Bifidobacterium* populations by ZEN at 290 ppb would increase *Enterobacter* and *Clostridium* abundances [52]. In our study, ZEN had no significant effect on SCFAs production in the colon, either at 75 ppb or 290 ppb concentrations, when compared to Control. By contrast, other studies reported a decrease in SCFAs overall under the influence of ZEN [7,9,13,16] while Liu et al. observed an increase in SCFAs [53]. Although not significant, a slight increase in the ammonia values could be seen in the presence of both ZEN concentrations in the diet. Any difference in pH concentration was registered irrespective of the treatment. Because pH guides the competition between different microbial species and pH differs in different colon segments, it was necessary to investigate this parameter, comparing it to physiologically healthy pH ranges within the colon [54]. The values obtained ranged from 6.69 in the Control group to 6.73 and 6.79 in ZEN 75 and ZEN 290, respectively, considered to be within the physiologically normal range [55]. Bacterial metabolism cooperates in producing a pH gradient throughout the colon from the proximal to the distal section [56]. With the absorption of SCFAs within the colon, the pH increases with cells also secreting bicarbonate [57]. Another factor contributing to the increase in pH is the subsequent release of ammonia and urea in the distal colon through protein and amino acids metabolisation. This process increases the pH close to 6.6 [54,56]. Ammonia is a toxic metabolite within the gastrointestinal tract of pigs resulting from amino acid deamination or urea hydrolysis [58]. It readily passes into the portal blood, metabolised by the liver into urea that is then eliminated in urine [59]. Ammonia concentration rests on the equilibrium derived from amino acid deamination on the one hand and microbial protein synthesis on the other hand [58]. Usually, an ammonia increase leads to the inhibition of growth and differentiation of intestinal epithelial cells [60], and thus impacts the colon health negatively as well as disturbing the bacterial balance within piglets’ gastrointestinal tract [61]. As mentioned above, no significant differences were observed in our study.

Furthermore, Pearson’s correlations suggest different relationships between microbiota and SCFAs production under both ZEN concentrations. Butyric acid was strongly correlated with the presence of *Lactobacillus* and *Bifidobacterium* within the ZEN 75 experimental group. Other studies have also mentioned a similar association [62,63]. *Enterobacter* was found to be moderately associated with both acetic acid and propionic acid in the ZEN 75 and 290 experimental groups. This association was also observed in several other trials [64,65] under differing experimental premises.

The changes induced by ZEN on bacterial abundances were also verified by clustering, using two PLS-DA analyses. While the microbial taxonomic and relative abundance composition for the ZEN 75 group was similar to those of the Control, both differed from the ZEN 290 group. The high concentration of ZEN (290 ppb) profoundly affected the relative abundances and bacterial composition, with an apparent distancing between the experimental groups being highlighted. It was also strongly associated with an increased presence of *Clostridium* and *Enterobacter* and with the slight differences in concentrations of acetic acid within the colon content while negatively associated with the *Lactobacillus* genera. These associations could also contribute to the apparition of diarrhoea symptoms and contribute to it. Many studies discussed the relationship between *Lactobacillus* and post-weaning diarrhoea [66,67,68] showing that lactic acid bacteria are frequently used as a probiotic in reducing diarrhoea incidence, as reported by Dell’Anno et al. [69] or Wang et al. [70]. Other studies also indicate increased diarrhoea incidence with decreased *Lactobacillus* counts [71,72].

In the gut, the microbiota interacts with the host’s immune system [73]. Secretory immunoglobulin A (sIgA) is the most abundant antibody isotype produced by B cells at mucosal surfaces as a response to the food antigens and the intestinal microbiota [74]. sIgA can neutralize toxins at the mucosal surfaces of the gut, protecting against pathogens and toxins [75]. In our experiment, the exposure of weaned piglets to zearalenone for 22 days induced a dose-dependent increase in sIgA in the colonic content, which was statistically significant in piglets exposed to the highest concentration of ZEN. Another recent study of Wang et al., [76] has similarly shown that the exposure to higher concentrations of ZEN (20 mg/kg b.w.), but for a shorter period (one week) can significantly induce an increase in faecal IgA levels. The sIgA synthesis is modulated by the complexity of gut microbiota and different bacterial populations can trigger an IgA response of different type and intensity [77]. On the other hand, sIgA can bind and coat the gut microorganisms controlling the microbiota composition, modulating the bacterial behaviours and enforcing host-microbiota homeostasis [78]. In the present study, the increase in sIgA could result in a defence response to the relative abundance of *Prevotella, Clostridium* and *Enterobacter* induced by ZEN. This could constitute a possible defence mechanism to regulate gut microbiota composition and to maintain gut homeostasis after toxin exposure by controlling the microorganisms’ mechanisms, reducing their inflammatory potential and facilitating their clearance from the gut [75,79].

Within the intestinal barrier, tight junctions are essential in forming selectively permeable seals between adjacent intestinal epithelial cells [18]. These seals are composed of transmembrane proteins (Claudins), TJ-associated marvel proteins (e.g., Occludin), junctional adhesion molecules (JAMs), and cytosolic proteins (e.g., Zonula). ZEN can damage these seals, decreasing intestinal permeability [20,24]. No significant alterations in the gene expressions for Claudin 4, ZO1, or Occludin could be observed within the colon tissue in the present study. These results obtained by qPCR were confirmed at the protein level by Western blot analysis, showing that neither the concentration of ZEN below the EC recommendation (75 ppb) nor that above (290 ppb) had any effect on these junction proteins. Zhang et al. in 2021 [27] reported similar results when ZEN at 150 ppb did not induce changes in gene and protein expression of Claudin 1, Occludin, or ZO1 junction proteins in weaned piglets. By contrast, a significant decrease in the expression of these proteins was observed only at increased concentrations of 1500 to 3000 ppb ZEN [27], leading to cecal physical barrier injury.

## 4. Conclusions

Weaned piglets’ exposure to contaminated feed with 75 ppb ZEN did not significantly impact the microbiota abundance or SCFA production within the colon, except for a significant decrease in *Lactobacillus* abundance. Nevertheless, this decrease could also be attributed to the microbiota changes when piglets change their diet after weaning. PLS-DA also shows that the differences between Control and ZEN 75 were not radical at a taxonomic level or relative abundance.

ZEN at 290 ppb also decreased the populations of *Lactobacillus* and *Bifidobacterium* significantly. In addition, it increased the relative abundance of *Clostridium*, *Enterobacter* and *Prevotella* compared to the Control. By contrast, there was a weak association between ZEN at 290 ppb and NH3 presence within the colon. PLS-DA analysis confirmed the existence of these changes at a microbiological level. ZEN 290 ppb also increased the percentage of diarrhoea incidence and IgA concentration (*p* < 0.05), while SCFA synthesis in the colon remained largely unaffected at both concentrations of ZEN.

As such, the presented findings bring new data regarding the effect of low concentrations of ZEN on young pigs. However, further analysis is needed to understand the underlying mechanisms that ZEN disrupts at the intestinal level.

## 5. Materials and Methods

### 5.1. Experimental Design

A total of 18 crossbred weaned piglets (TOPIGS-40) were used for the in vivo experiment and were randomly assigned to three groups (6 animals/group) as follows: (1) control group (Control) fed uncontaminated feed; (2) group ZEN 75 fed a diet contaminated with 75 ppb ZEN, and (3) group ZEN 290 fed a diet contaminated with 290 ppb ZEN. The diets were designed to meet the requirements of NRC 2012 weaned piglets (Table 4). Piglets were individually ear-tagged, and every piglet was considered an experimental unit, with six experimental units per dietary group. Each group was housed in a separate pen. During the experimental trial, diarrhoea was recorded daily for each piglet. The number of days with diarrhoea was counted for each experimental group and expressed as a percentage of the total experimental period (22 days). Before ingesting the mycotoxin-contaminated diets, weaned piglets were acclimatized for one week. The young piglets were clinically normal, and no deaths resulted from the experiment. During the experiment, piglets had free access to solid feed and water. After 22 days, all animals (6 animals/group) were electrically stunned and exsanguinated (according to the 2010/63/EC directive of the EU Council). Colon and colonic content samples were collected from each animal and stored at −80 °C until further analyses. The handling and protection of the animals used in this experiment were realized in compliance with the 206/2004 Romanian law and the 98/58/EC Council Directive of the EU Council concerning treating animals used for experimental purposes.

### 5.2. Toxins

ZEN (>98% pure by high-performance liquid chromatography [HPLC]) used to contaminate the experimental diets was purchased from FERMENTEC (Jerusalem, Israel) and dissolved in DMSO (dimethyl sulfoxide): water (1:7) solution and then mixed into the basal diet in order to provide a feed containing 75 µg/kg 75 ppb and 290 µg/kg 290 ppb concentration confirmed by ELISA analysis. Experimental diets were screened by ELISA for other mycotoxins (aflatoxin B1, deoxynivalenol, ochratoxin A, fumonisins, and toxin T2) using ELISA kits Veratox (Neogen, Lansing, MI, USA) according to the manufacturer’s instructions. Their concentrations were under the detection limits of the kits.

### 5.3. Microbial DNA Extraction

A QIAamp DNA stool minikit extraction kit (QIAGEN, Dusseldorf, Germany) was used for microbial genetic material extraction. A previous study already described primer sequences (Eurogentec, Seraing, Belgium) and the protocol of PCR amplification of universal 16S regions [80]. Standard curves were generated using serial dilutions of template DNA (105–1012 molecules/µL). The selective bacteria genera DNA was quantified using the RotorGene Q series PCR system (Qiagen, Hilden, Germany) with species-specific primers sequences designed for the following genera: *Lactobacillus*, *Clostridium*, *Enterobacter*, *Prevotella* and *Bifidobacterium* and using a PCR protocol already described elsewhere [80,81]. The gene copy numbers for the selected microorganisms were obtained using the Cq values relative to the standard curve. The final copy number for each targeted bacteria genus was calculated using the following equation:*No. bacterial copies digesta/grams = (CM × AC × DV)/q × S*
in which *CM* represents the quantitative mean of the copy number, *AC* represents the DNA concentration of each sample, *DV* is the dilution volume of extracted DNA, *q* is the DNA amount (ng) which was used in the experiment, and *S* stands for the weight of the colonic sample (g) [62].

### 5.4. Determination of SCFAs, Ammonia Content and pH Value

A volume of 25 mL of digesta from the colon was used for pH analysis using a digital pH meter (Dostmann electronic, GmbH, Wertheim, Germany) following the manufacturer’s protocol [82]. The ammonia concentration was measured using an ammonia selective-ion electrode (Orion 95–12, Orion Research Inc., Franklin, MA, USA) attached to a selective ion meter (710A model, Orion Research Inc., Franklin, MA, USA). The calibration was performed using dilutions at 10, 100, and 1000 mg/L of a 0.1 M standard ammonium chloride solution. In addition, an ISA solution (Ionic Strength Adjuster, Orion Research Inc., Franklin, MA, USA) of 0.5 mL was mixed into every 25 mL sample to ensure a uniform background ionic strength [83] The final results were reported as µmol/g.

Short-chain fatty acids from the piglet colonic digesta samples were analysed in aqueous extracts by gas chromatography as described by Marin et al. [80]. Briefly, the aqueous extracts were injected into a gas chromatograph (Varian, 430-GC, Varian Inc., Palo Alto, CA, USA) after centrifugation. The GC instrument is fitted with an Elite-FFAP capillary column (inside diameter 320 mm) (Perkin Elmer, Waltham, MA, USA). Volatile fatty acids obtained from CRM46975, Supelco, Bellefonte, PA, USA, were used as standards. The final results were reported as µmol/g.

### 5.5. Quantification of Gene Expression by qPCR

The effects produced by the experimental diets on genes encoding for tight junction markers Claudin 4 (CLDN4), Zonula1 (ZO1), and Occludin (OCCLDN) were determined by qPCR. Methods cited in Pistol et al., 2021 were used to extract total ARN, the transcription into complementary DNA (cDNA) and the quantitative PCR [84]. The Excel-based NormFinder software was utilised to normalise qPCR data and reference genes selected from 6 housekeeping genes analysed. The 2(−ΔΔC)q 2^(−ΔΔCt)^ formula was used for the comparison to the control group, thus obtaining the Fold Change.

### 5.6. Detection of Mucosal IgA in the Colonic Content

Lyophilised colonic content (0.25 g) was first diluted in 1 mL PBS and incubated at 40 °C for ten minutes, followed by a five minutes centrifugation at 5590G. Next, a 1:2 dilution in a solution of glycerol 40% in PBS was used for the supernatant. The IgA concentration in the samples was analysed using a pig IgA ELISA kit (Bethyl Laboratories Montgomery, TX, USA) according to the manufacturer’s instructions. Next, the absorbance was read at 450 nm using a plate multi-reader (Varioscan, Thermo Scientific, Waltham, MA, USA), and the IgA content (ng/mL) was calculated.

### 5.7. Immunoblotting Analysis

Protein expression of three tight junction proteins: Claudin 4 (CLDN4), Zonula 1 (ZO1), and Occludin (OCCLDN), was determined by Western blot. The colon samples (50 mg) were homogenised in RIPA buffer. A Pierce BCA Protein Assay Kit, Thermo Fischer Scientific, USA, was used to quantify the protein content within the obtained lysates. A 10% SDS-PAGE electrophoresis was performed, and 30 µg of proteins were moved to the nitrocellulose membrane, blocked and incubated with specific primary and secondary antibodies as described by Pistol et al., 2021 [84]. The MicroChemi Imager (DNR Bio-Imaging Systems LTD, Neve Yamin, Israel) was used to develop the immunoblotting images, and the GelQuant software (DNR Bio-Imaging Systems LTD, Neve Yamin, Israel) was utilised to assay the protein expression level. A ratio of the expression levels between the protein of interest and β-actin was calculated to express the results.

### 5.8. Statistical Analyses

Results are presented as mean ± SEM. XLSTAT (Addinsoft, Paris, France) was used to perform a one-way ANOVA analysis and derive significance between the three experimental groups and Dennett’s and Tukey’s HSD tests. Statistical significance was considered for *p* values < 0.05 (* *p* < 0.05, ** *p* < 0.01, **** *p* < 0.0001). Partial least square—discriminant analysis was performed using XLSTAT (Addinsoft, Paris, France) to analyse clustering occurrence and the separation magnitude among the experimental groups and variables. Pearson’s correlation coefficients between SCFA concentration and microbiota in the colon content as well as associated heatmaps, were calculated using XLSTAT (Addinsoft, Paris, France).

## Figures and Tables

**Figure 1 toxins-15-00206-f001:**
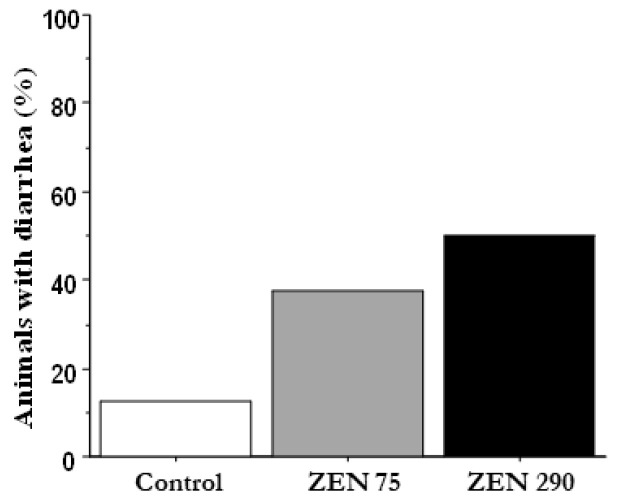
Percentage of animals with diarrhoea after the exposure to zearalenone.

**Figure 2 toxins-15-00206-f002:**
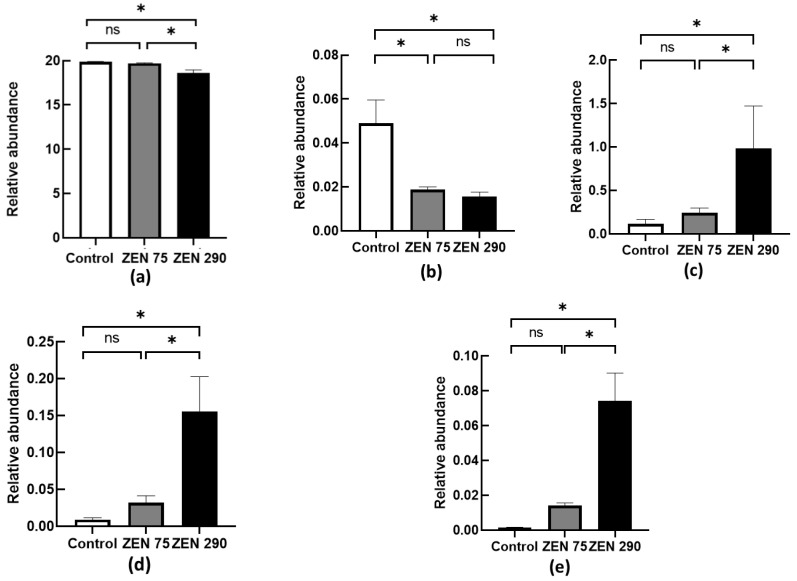
The effect of ZEN on the relative abundance of important gut bacteria populations: (**a**) *Bifidobacterium,* (**b**) *Lactobacillus,* (**c**) *Prevotella,* (**d**) *Clostridium,* (**e**) *Enterobacter.* The results are displayed as mean ± SEM. * indicates a significant difference (*p* < 0.05); “ns” indicates no significance (*p* > 0.05).

**Figure 3 toxins-15-00206-f003:**
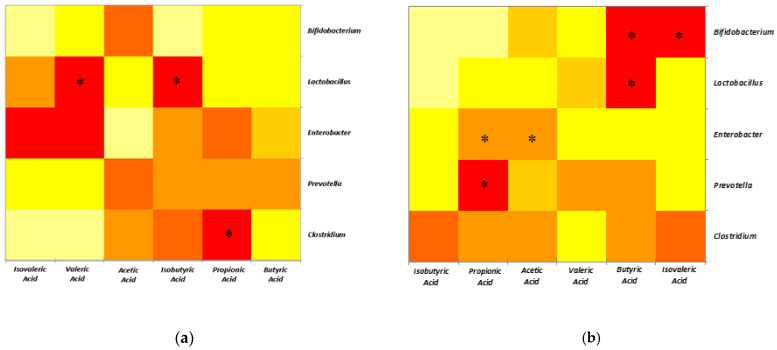
Heatmap of correlations between microorganism populations and SCFAs corresponding to Control (**a**), ZEN at 75 ppb (**b**), and ZEN at 290 ppb (**c**). Red indicates positive correlation, yellow stands for a negative correlation. * indicates the statistical significance attached to the correlation (*p* < 0.05).

**Figure 4 toxins-15-00206-f004:**
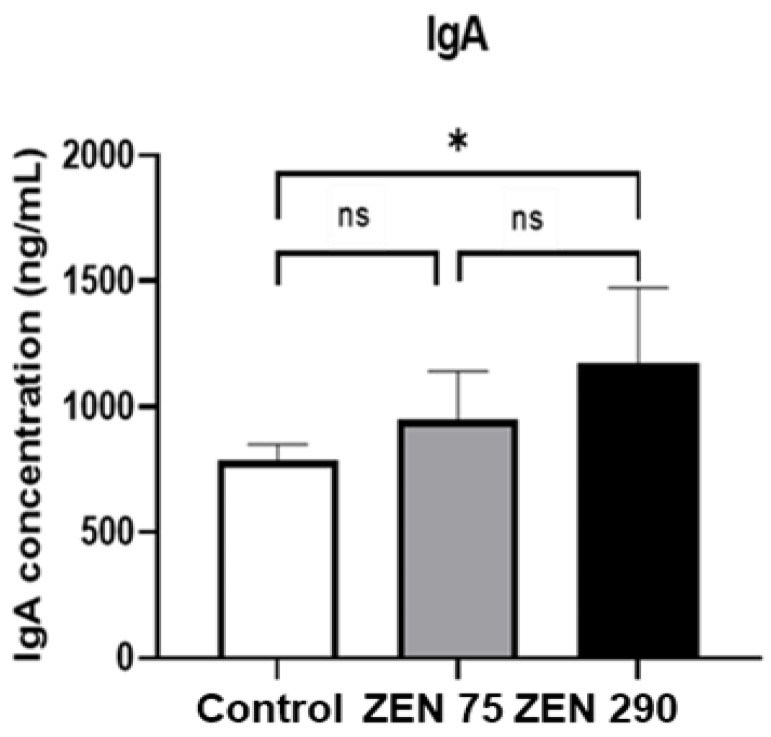
Effect of ZEN on IgA secretion in the colonic content of the weaned piglets. The results are displayed as mean ± SEM. * indicates a significant difference (*p* < 0.05).

**Figure 5 toxins-15-00206-f005:**
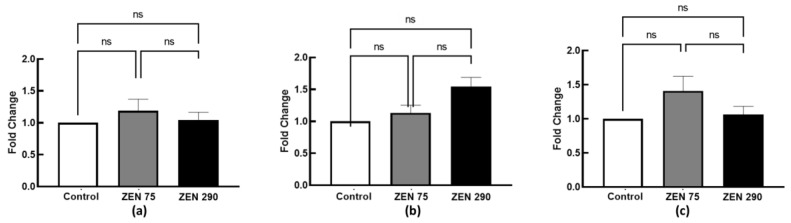
Fold change in gene expressions corresponding to (**a**) *Cldn 4*, (**b**) *ZO1,* and (**c**) *Occldn* junction proteins. The results are displayed as mean ± SEM.

**Figure 6 toxins-15-00206-f006:**
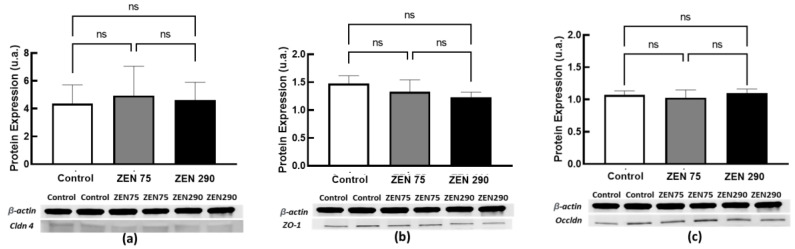
Protein expression of junction proteins (**a**) *Cldn 4*, (**b**) *ZO1*, and (**c**) *Occldn*. The results are displayed as mean ± SEM.

**Figure 7 toxins-15-00206-f007:**
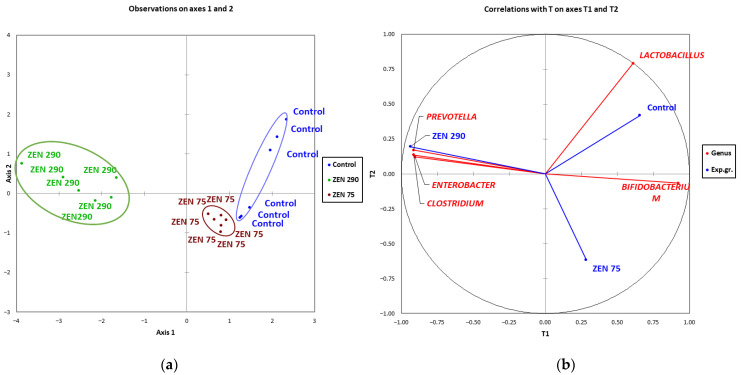
The relationship between the ZEN effect and selected bacterial genera in the experimental groups using partial least squares discriminant analysis (PLS-DA). (**a**) Score plot for observations with each point representing a sample, points of the same colour belonging to the same experimental group, and points of the same group are marked with ellipses (**b**) Correlation plot between genus and experimental groups.

**Figure 8 toxins-15-00206-f008:**
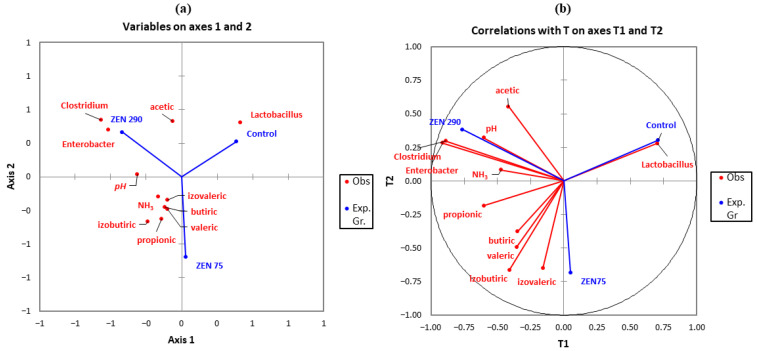
The effect of ZEN on the other analysed parameters (selected bacterial species, SCFAs, pH and ammonia concentration) using Partial Least Squares Discriminant Analysis (PLS-DA). Discriminating between (**a**) experimental groups and selected parameters via weights plot (**b**) between genus and experimental groups by a correlation plot.

**Table 1 toxins-15-00206-t001:** The effect of the diets on the performance of piglets.

Parameters *	Control	ZEN 75	ZEN 290	*p* Value
Initial BW (kg)	11.7 ± 0.9	11 ± 1	11 ± 1.4	0.577
Final BW (kg)	21.5 ± 2.8	19.6 ± 3.5	19.5 ± 1	0.465
Average Daily Gain (kg)	0.42 ± 0.1	0.3 ± 0.1	0.3 ± 0.1	0.656
Average Daily Feed Intake (kg)	1.1 ± 0.2	1.0 ± 0.2	1.0 ± 0.2	0.177
Feed Efficiency (kg/kg)	2.6 ± 0.2	2.7 ± 0.3	2.8 ± 0.3	0.988

* Results are represented as the mean ± SEM.

**Table 2 toxins-15-00206-t002:** SCFA concentrations (mM/L), NH_3_ concentrations (µM/g) and pH values from colonic content of piglets fed ZEN 75 and ZEN 290 ppb.

SCFA (mM/L) /Experimental Group *	Acetic Acid (mM/L)	Propionic Acid (mM/L)	Isobutyric Acid (mM/L)	Butyric Acid (mM/L)	Isovaleric Acid (mM/L)	Valeric Acid (mM/L)	Total SCFAs (mM/L)	pH	NH_3_ (µM/g)
Control	22.58 ± 0.6	7.45 ± 0.4	0.67 ± 0.1	4.35 ± 0.3	0.70 ± 0.1	1.14 ± 0.1	38.67 ± 1.3	6.69 ± 0.0	11.10 ± 0.9
ZEN 75	19.50 ± 1.9	8.89 ± 0.8	0.90 ± 0.1	5.33 ± 0.8	0.86 ± 0.2	1.40 ± 0.3	39.46 ± 3.8	6.73 ± 0.0	12.70 ± 1.9
ZEN 290	23.42 ± 2.6	8.60 ± 1.1	0.88 ± 0.1	5.15 ± 0.9	0.84 ± 0.1	1.33 ± 0.2	42.52 ± 4.4	6.78 ± 0.0	13.07 ± 0.7

* Results are represented as the mean ± SEM.

**Table 3 toxins-15-00206-t003:** Pearson’s correlation between bacterial populations in colon digesta and short-chain fatty acid production corresponding to Control, ZEN 75, and ZEN 290 experimental groups.

Experimental Group	SCFA/ Bacterial Genus	Acetic Acid	Propionic Acid	Isobutyric Acid	Butyric Acid	Isovaleric Acid	Valeric Acid
Control	*Clostridium*	0.434	0.723 (*p* = 0.0288)	0.553	−0.015	−0.444	−0.438
	*Prevotella*	0.469	0.378	0.361	0.317	−0.096	−0.074
	*Enterobacter*	−0.371	0.538	0.426	0.250	0.727	0.724
	*Lactobacillus*	−0.070	−0.006	0.682 (*p* = 0.0399)	0.075	0.422	0.724 (*p* = 0.0214)
	*Bifidobacterium*	0.570	0.076	−0.366	0.094	−0.630	−0.114
ZEN 75	*Clostridium*	0.470	0.430	0.520	0.451	0.515	0.313
	*Prevotella*	0.410	0.680 (*p* = 0.0156)	0.343	0.481	0.307	0.485
	*Enterobacter*	0.430 (*p* = 0.0439)	0.435 (*p* = 0.0346)	0.262	0.256	0.230	0.196
	*Lactobacillus*	0.244	0.223	0.074	0.755 (*p* = 0.0353)	0.220	0.420
	*Bifidobacterium*	0.394	0.112	0.100	0.744 (*p* = 0.0180)	0.744 (*p* = 0.0189)	0.225
ZEN 290	*Clostridium*	0.370	0.581	0.265	0.120	0.246	−0.482
	*Prevotella*	0.480	0.620	0.277	0.650	0.400	0.574
	*Enterobacter*	0.630 (*p* = 0.0144)	0.660 (*p* = 0.0416)	0.149	0.140	0.157	0.193
	*Lactobacillus*	−0.770 (*p* = 0.0267)	−0.446	−0.410	0.652 (*p* = 0.0155)	−0.482	0.539
	*Bifidobacterium*	−0.800 (*p* = 0.0124)	−0.640 (*p* = 0.0302)	−0.420	0.700 (*p* = 0.0244)	0.295	0.380

**Table 4 toxins-15-00206-t004:** Composition and nutrient content of experimental diets (%).

Ingredients (%)	Control	ZEN 75	ZEN 290
Corn	68.46	68.46	68.46
Soybean meal	19.00	19.00	19.00
Milk replacement	5.00	5.00	5.00
Corn gluten	4.00	4.00	4.00
l-Lysine	0.31	0.31	0.31
Methionine	0.10	0.10	0.10
CaCo_3_	1.57	1.57	1.57
Ca(H_2_PO_4_)_2_	0.35	0.35	0.35
NaCl	0.10	0.10	0.10
Mineral-Vitamin Premix	1.00	1.00	1.00
Choline Premix	0.10	0.10	0.10
Phytase	0.01	0.01	0.01
ZEN (mg/kg)	-	75	290
Metabolizable Energy (kcal/kg)	3282.60	3282.60	3282.60
Crude protein %	18.70	18.70	18.70
Lysine %	1.20	1.20	1.20
(Methionine + Cysteine) %	0.72	0.72	0.72
Ca %	0.90	0.90	0.90
P %	0.72	0.72	0.72
Fats %	2.40	2.40	2.40
Cellulose %	4.24	4.24	4.24
Dry Matter %	89.44	89.44	89.44

## Data Availability

The analysed sets of data used in the present paper can be offered, on reasonable request, by the corresponding author.

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
