# Peer review of "Effects of Exposure to Low Zearalenone Concentrations Close to the EU Recommended Value on Weaned Piglets’ Colon"

_toxins, 2023, doi:10.3390/toxins15030206_

Round 1
Reviewer 1 Report
OVERALL EVALUATION
The manuscript, although not very original, is potentially interesting, unfortunately too many important parameters (pH, NH3, lactic acid) are missing to understand the effect of zearalenone on the intestinal microflora. Furthermore, there is a lack of insight into the link between the health status of animals (eg. diarrhea) and the bacterial composition of intestinal microflora. The manuscript can not be accepted in the present form and need a major revision.
MAJOR REMARKS
· Figure 1. It is not reported whether the differences are statistically significant or not. · Were the animals reared in single or group cages? What was the experimental unit, the animal or the cage?· Table 2. Authors should add a column with the total VFA production for each treatment.
· Table 2. The concentration of lactic acid is not reported, although there is a clear effect of ZEA on Lactobacillus.· Table 2. Other important parameters, not included in Table 2, are pH and NH3.
· Table 3. In addition to Pearson's r value, P value must also be indicated, otherwise it is not clear when a relationship is significant and when it is not.
· Were the incidence and severity of diarrhea related to the presence of Lactobacilli or other groups (Clostridia ?) or to pH or single VFA production?
Author Response
We would like to thank the Reviewer 1 for pointing out the missing information and for helping us in bridging the gaps in order to improve our paper.
A point by point response follows.
- Comment: Figure 1. It is not reported whether the differences are statistically significant or not.Were the animals reared in single or group cages? What was the experimental unit, the animal or the cage?
Authors Response:
Thank you for your valuable observation. The animals were reared in pens, 6 animals per pen, one pen for each experimental group. Each animal was considered to be an experimental unit This is added in the new version of the manuscript (Material and methods, 5.1 Experimental design, lines 420-422). During the experimental trial, the occurrence of diarrhoea was checked daily for each piglet and the number of the days with diarrhoea were counted for each experimental group and expressed as percentage from the total number of experimental period (22 days). For this reason, we were not able to perform any statistical analysis. All these information was added in the new version of the manuscript (5.1 Experimental design, lines 420-427):
- Comment: Table 2. Authors should add a column with the total VFA production for each treatment.
Authors Response:
As recommended we included a column with the total VFA production for each treatment (lines 154-156). A supplementary commentary was also added in the Results chapter lines 148-150
- Comment: Table 2. The concentration of lactic acid is not reported, although there is a clear effect of ZEA on Lactobacillus.
Authors Response:
Thank you for this suitable observation. Although not a short-chain fatty acid, lactate is a metabolite produced by gut bacteria through similar processes and metabolic pathways as SCFAs as described in other studies (Tan J. et al., 2014, Markowiak-Kopeć P. et al., 2020) and makes valuable contributions to the health of pigs.
We are indeed confident that the lactate concentrations were affected, with the population decrease of Lactobacillus, which was shown to be inversely correlated to the ZEN presence in the diets through a new PLS-DA analysis (also relevant to the point 6 that was raised). That is why we wanted to analyze the concentration of lactic acid, but we could not because of laboratory technical limitation. We do not have this method among our optimized techniques.
- Comment: Table 2. Other important parameters, not included in Table 2, are pH and NH3.
Authors Response:
Thank you for your suggestion. These parameters were included in Table 2 and further commented in Results chapter, lines 154-156 as well as in Discussions chapter lines 321-347 and the analysis methodology was added in Materials and Methods Chapter, lines 463-472.
- Comment: Table 3. In addition to Pearson's r value, P value must also be indicated, otherwise it is not clear when a relationship is significant and when it is not.
Authors Response:
Thank you for this pertinent observation. p values were calculated for all correlations done and were presented in the modified Table 3 in cases of significance. This also changed the relevant obtained correlations as well as producing new cases of correlation therefore supplementary commentaries were added and significance commented in the lines 160-181 of Results chapter and in the lines 342-348 for the Discussions chapter.
- Comment: Were the incidence and severity of diarrhea related to the presence of Lactobacilli or other groups (Clostridia ?) or to pH or single VFA production?
Authors Response:
Thank you for raising attention to this issue. In order to better understand this issue we have performed a new PLS-DA analysis which takes into account the variables that the reviewer requested and running them through a vector axis represented by the experimental groups with ZEN at 75 and 290 ppb as well as for the Control group.
There was a strong visible association between ZEN at 75ppb and isobutiric and isovaleric acids as well as between ZEN at 290ppb and the increase in Clostridium and Enterobacter populations on one hand and acetic acid on the other, as well as being moderately associated with the NH3 concentrations. These findings were also commented in the manuscript in the Results chapter at 237-244 lines as well as in the Discussion chapter at 354-362 lines.
The entire manuscript was reedited, grammar, spelling and structure changed and the readability was improved.

Reviewer 2 Report
The manuscript aims to evaluate the potential adverse effect of zearalenone (to be abbreviated ZEN according to international guidelines) on intestinal integrity, sIgA production, colon microbiota and SCFA production. The authors elected two low concentrations of ZEN, which makes the approach attractive as in previous investigations often rather high toxin concentrations were used, which are only incidentally found in daily practice. The authors are invited to consider the following general comments:
Please provide a complete description of the diet (all components).
Please provide a clear description of the source of ZEN. No detailed description of the diet is given.
Please provide a transparent description of the clinical status. The pigs had a varying incidence of diarrhoea; hence a clear diagnostic workout should be given to clarify the origin of diarrhoea, as circumstantial bacterial infection could have influenced the results. Was this diarrhoea or only loose faeces?
The abundancy Clostridium spp and Enterobacteriaceae was increased dose dependently. Please address thus more clearly in the discussion section, as both general include well-knowns pathogens affecting the intestinal health of pigs (was there a correlation between such shifts in microbiota and clinically visible diarrhoea?)
The authors state (line 130-134, 261): “In the present study, out of all the SCFAs, acetic acid was the most abundant short chain fatty acid, followed by propionic acid”….. what does this mean, as the difference between control and ZEN groups are non-significant. In contrast, there are some trends I increase levels pf butyric acid and isobutyric acid, which are worthwhile to be included in the discussion.
Table 3 / Figure 3: Why a Pearson correlation is not presented for the control group (please add)
Figure 4 shows and dose-dependent increase in the sIgA concentration in the colonic content. Generally, this would be considered as a beneficial effect in gut health monitoring. Please comment.
Considering their working hypothesis, the authors are invited to include (ate least in the discussion section) a justification of the selected parameters and explain why – for example - no markers of inflammation were included.
The authors are encouraged to involve a native speaker for editing of the text and even the title. Many parts of the manuscript are repetitive and could be better structured and considerably shortened.
Editorials:
Please replace the abbreviation ZEA by ZEN (the more recently recommended abbreviation).
Please replace CE by EC in the entire manuscript.
Line 7: please use only mg/kg or µg/kg to describe toxin concentrations in feed materials.
Line 17-19: please replace this statement by a more neutrals sentence such as for example: the presented findings contribute to a better understanding of the dose-dependent adverse effect of ZEN in the intestinal tract of pigs. This applies also to the text in lines
Line 45 rephrase: short chain fatty acids (SCFA)
Line 57 to 59: the studies cited here refer to experiments with very high ZEN concentrations and not very helpful in this context. Please stress this difference.
Line 66: please add the reasons for the high sensitivity of pigs to ZEN – referring to the hepatic biotransformation yielding alpha-ZOL for example.
Line 73: rephase tight junction “strands” by tight junction network – or explain.
Line 174. Heading – should read: the effect of dietary zearalenone on tight junction proteins.
Line 333 – 336: Please delete these sentences as they are non-justified (see also comment above regarding the last sentence of the abstract). EEFSA (EC) risk assessment is based on a comprehensive review and evaluation of all available data and relevant endpoints and hence a single article cannot claim to make a major contribution to EC regulations.
Author Response
We would like to thank the Reviewer 2 for pointing out the missing information and for helping us in bridging the gaps in order to improve our paper.
A point by point response follows.
- Please provide a complete description of the diet (all components).
Authors Response:
The composition of the experimental diet and nutrients is provided in the material and methods of the new version of the manuscript (5.1 Experimental Design (lines 434-435 of the revised manuscript).
- Please provide a clear description of the source of ZEN. No detailed description of the diet is given.
Authors Response:
The source of ZEN mycotoxin is provided in material and method (subchapter 5.2 Toxins) in the new version of manuscript. The concentrations used with the experimental diets were also included (for the previous point raised) (438-445).
- Please provide a transparent description of the clinical status. The pigs had a varying incidence of diarrhoea; hence a clear diagnostic workout should be given to clarify the origin of diarrhoea, as circumstantial bacterial infection could have influenced the results. Was this diarrhoea or only loose faeces?
Authors Response:
Thank you for your comment. At the beginning of the experiment all the piglets were healthy and they were randomly attributed to different experimental groups. The diarrhoea cases appear after feeding the animals with experimental diets and were directlly proportional with the concentration of ZEN in the diet. Piglets with diarrhoea was sporadic and should be rather correlated with the ingesta of contaminated feed rather than with a bacterial infection that would be affected all the piglets from the same box/treatement, including the control. If a bacterial infection appeared, this would affected all the piglets in a group.
Regarding diarrhea consistence, the altered faeces reported in the paper had a semi-liquid consistency.
- The abundancy Clostridium spp and Enterobacteriaceae was increased dose dependently. Please address thus more clearly in the discussion section, as both general include well-knowns pathogens affecting the intestinal health of pigs (was there a correlation between such shifts in microbiota and clinically visible diarrhoea?)
Authors Response:
Thank you for your valuable observation. Indeed, Clostridium and Enterobacter were highly abundant in the colon of piglets fed the diet contaminated with 290ppb of ZEN and the incidence of diarrhea was significantly higher in these piglets compared to the control piglets and those that were contaminated with ZEN 75 ppb.
A supplementary discussion was provided within the text (lines 277-290). Although the Clostridium and Enterobacter genera include well known pathogens, we have to take into account that many species of these genera are also normal members of a healthy pig microbiota as pointed out by many other studies.
- The authors state (line 130-134, 261): “In the present study, out of all the SCFAs, acetic acid was the most abundant short chain fatty acid, followed by propionic acid”….. what does this mean, as the difference between control and ZEN groups are non-significant. In contrast, there are some trends I increase levels pf butyric acid and isobutyric acid, which are worthwhile to be included in the discussion.
Authors Response:
We are sorry for this misunderstanding. We meant to say that out of all the short chained fatty acids analysed in our experiment, acetic acid was found in the largest quantity, no matter of the experimental group. We have reformulated our phrase, in order to avoid the misunderstanding and we have included it in the new version of the manuscript at the lines 311-314
Regarding the second part of the Reviewer comment, no discussion was made concerning the increased levels of butyric acid and isobutyric acid in ZEN groups as reported to control group, as these differences were no significant (P<0.05) or showed a trend to be significant (P<0.1). [(P values for butyric acid: P=0.297 (ZEN 75 vs Control); P=0.612 (ZEN 290 vs Control); for isobutyric acid: P=0.156 (ZEN 75 vs Control); P=0.28 (ZEN 290 vs Control)].
- Table 3 / Figure 3: Why a Pearson correlation is not presented for the control group (please add)
Authors Response:
As recommended, Pearson correlations as well as heatmaps were provided in the new version of the manuscript for the control group as well as relevant commentary as specified (Table 3 revised and Figure 3 revised lines 182-192)
- Figure 4 shows and dose-dependent increase in the sIgA concentration in the colonic content. Generally, this would be considered as a beneficial effect in gut health monitoring. Please comment.
Authors Response:
Thank you for this comment. Indeed, an increase in secretory Immunoglobulin A concentration has a beneficial effect for gut health. Specifically, the intestinal epithelial cells (IECS), mucus, sIgA and antimicrobial peptides (AMPs) constitute a mucosal defense network, which protect the body from microbial flora, viruses and environmental pollutants. Mucosal sIgA not only protects intestinal epithelial cells from environmental toxins and pathogenic bacteria but also maintains the homeostasis of the intestinal mucosal environment. An increase of sIgA concentration can represent the response of the gut to a xenobiotic (in this case zearalenone) in order to maintain the gut homeostasis after toxin exposure . Several studies have shown that sIgA are responsible for toxins neutralisation, regulation of gut microbiota composition, decrese of their inflammatory potential and facilitation of their clearance from the gut (Mantis et al., 2011; Leon and Francino, 2022), in an atempt to counteract the negative effects associated with the toxins. Similar results were obtained by other researchers (Wang et al.,2018) that have shown that the exposure of mice to higher concentrations of ZEN (20 mg/kg b.w.) but for a shorter period (one week) can significantly induce an increase of faecal IgA levels. However, a longer exposure to toxins could maybe result in a decrease of sIgA synthesis with severe implications for the gut health.
- Considering their working hypothesis, the authors are invited to include (ate least in the discussion section) a justification of the selected parameters and explain why – for example - no markers of inflammation were included.
Authors Response:
As recommended, we included in the discussion section of the new version of the manuscript the explanation for which the current investigated parameters were chosen (lines 261-269). The effect of ZEN on inflammatory markers were indeed analysed and found not statistically significant.
Please find below these results:
|
Inflammatory markers |
Control (pg/ml) |
ZEN 75 (pg/ml) |
ZEN 290 (pg/ml) |
p Values |
||
|
|
|
|
|
Control Vs ZEN 75 |
Control Vs ZEN 290 |
ZEN 75 vs ZEN 290 |
|
IFN-γ |
717.32±229.7 |
635.26±127.9 |
1060.97±499.9 |
0.9796 |
0.6659 |
0.5494 |
|
IL 8 |
1980.69±209.8 |
1699.19±134.1 |
2279.12±116.7 |
0.5066 |
0.4682 |
0.0856 |
|
TNF-α |
398.47±55.5 |
411.60±27.4 |
390.12±32.4 |
0.976 |
0.9902 |
0.9372 |
- The authors are encouraged to involve a native speaker for editing of the text and even the title. Many parts of the manuscript are repetitive and could be better structured and considerably shortened.
Authors Response:
Thank you for your suggestion. The entire manuscript was reedited, grammar, spelling and structure changed and the readability was improved.
For Editorials:
- Please replace the abbreviation ZEA by ZEN (the more recently recommended abbreviation).
Authors Response:
The replacement was made throughout the paper
- Please replace CE by EC in the entire manuscript.
Authors Response:
The change was effectuated throughout the paper
- Line 7: please use only mg/kg or µg/kg to describe toxin concentrations in feed materials.
Authors Response:
Thank you. The suggestion was implemented throughout the manuscript
- Line 17-19: please replace this statement by a more neutrals sentence such as for example: the presented findings contribute to a better understanding of the dose-dependent adverse effect of ZEN in the intestinal tract of pigs. This applies also to the text in lines
Authors Response:
As suggested, the statement was modified accordingly and is now found in the revised form of the manuscript lines 17-18
- Line 45 rephrase: short chain fatty acids (SCFA)
Authors Response:
The correction was made throughout the manuscript
- Line 57 to 59: the studies cited here refer to experiments with very high ZEN concentrations and not very helpful in this context. Please stress this difference.
Authors Response:
Thank you for pointing out this difference. New comments were added to help explain the differences in ZEN concentrations used in our experiment and the ZEN concentrations used by the cited authors (lines 57-59 and 61-63)
- Line 66: please add the reasons for the high sensitivity of pigs to ZEN – referring to the hepatic biotransformation yielding alpha-ZOL for example.
Authors Response:
Thank you for raising this valid observation. The reasons behind the high sensitivity of pigs to ZEN were added (lines 69-76).
- Line 73: rephase tight junction “strands” by tight junction network – or explain.
Authors Response:
The correction was made (line 81).
- Line 174. Heading – should read: the effect of dietary zearalenone on tight junction proteins.
Authors Response:
The heading was modified accordingly (line 202)
- Line 333 – 336: Please delete these sentences as they are non-justified (see also comment above regarding the last sentence of the abstract). EEFSA (EC) risk assessment is based on a comprehensive review and evaluation of all available data and relevant endpoints and hence a single article cannot claim to make a major contribution to EC regulations.
Authors Response:
Thank you, the suggested line was removed

Round 2
Reviewer 2 Report
The authors have improved the scientific content of the manuscript including the comments of this referee. A major concern remains the non-satisfying and frustrating presentation of the introduction (jumping between issues) and the discussion section (repeating too many parts of the result section almost unchanged).
In the copy, sentences that absolutely require editing, (marked in yellow), should be deleted (red), or replaced (grey) are indicated. Please note that these recommendations for changes are limited to the essential parts and non-exhaustive.
Author Response
Thank you for your valuable observations. As recommended we edited the Introduction for more a suitable readability and increased comprehension of this section as well as regarding the discussion section
